# Dietary Anti-Aging Polyphenols and Potential Mechanisms

**DOI:** 10.3390/antiox10020283

**Published:** 2021-02-13

**Authors:** Jing Luo, Hongwei Si, Zhenquan Jia, Dongmin Liu

**Affiliations:** 1Guangdong Provincial Key Laboratory of Food, Nutrition and Health, Department of Nutrition, School of Public Health, Sun Yat-sen University, Guangzhou, Guangdong 510080, China; luojing27@mail.sysu.edu.cn; 2Department of Human Sciences, Tennessee State University, Nashville, TN 37209, USA; hsi@tnstate.edu; 3Department of Biology, University of North Carolina, Greensboro, NC 27402, USA; z_jia@uncg.edu; 4Department of Human Nutrition, Foods and Exercise, College of Agricultural and Life Sciences, Virginia Tech, Blacksburg, VA 24061, USA

**Keywords:** polyphenol, aging, antioxidant, cellular senescence

## Abstract

For years, the consumption of a diet rich in fruits and vegetables has been considered healthy, increasing longevity, and decreasing morbidities. With the assistance of basic research investigating the potential mechanisms, it has become clear that the beneficial effects of plant-based foods are mainly due to the large amount of bioactive phenolic compounds contained. Indeed, substantial dietary intervention studies in humans have supported that the supplementation of polyphenols have various health-promoting effects, especially in the elderly population. In vitro examinations on the anti-aging mechanisms of polyphenols have been widely performed, using different types of natural and synthetic phenolic compounds. The aim of this review is to critically evaluate the experimental evidence demonstrating the beneficial effects of polyphenols on aging-related diseases. We highlight the potential anti-aging mechanisms of polyphenols, including antioxidant signaling, preventing cellular senescence, targeting microRNA, influencing NO bioavailability, and promoting mitochondrial function. While the trends on utilizing polyphenols in preventing aging-related disorders are getting growing attention, we suggest the exploration of the beneficial effects of the combination of multiple polyphenols or polyphenol-rich foods, as this would be more physiologically relevant to daily life.

## 1. Introduction

The common understanding of food has evolved from the previous concept “eat to live” to health promotion and disease prevention. Substantial dietary intervention studies and epidemiological surveys have demonstrated that the consumption of a diet rich in fruits and vegetables has health-promoting effects [1,2]. It is estimated that the proportion of people aged 60 and above will increase from 12% in 2015 to 22% in 2050 [3]. Thus, delaying or preventing the onset of aging-related diseases due to cellular damage and functional decline would greatly improve life quality and expectancy, as well as mitigate the burden on the current healthcare system.

Despite the debate on the free radical theory of aging for the past decade [4], it is widely acknowledged that disrupted homeostasis between oxidants and antioxidants would cause cellular damage and even organ dysfunction [5]. To counteract the abnormal oxidative stress and accumulation of excess oxidants such as reactive oxygen species (ROS), which cause DNA damage [6] and cellular senescence [7], antioxidants bind to the pro-oxidants and abstract hydrogen to form stabilized radicals, hence neutralizing the hazardous effects of oxidants [8]. Oxidative stress plays an important role in aging process and various aging-associated chronic diseases. Reducing ROS has been shown to alleviate oxidative damage and extend the lifespan of various animal species [9]. Thus, phytonutrients with antioxidant property have drawn great attention for their ROS-scavenging actions and anti-aging potential.

Polyphenols are the largest, most studied group of naturally occurring antioxidants, which can be structurally categorized into phenolic acids, flavonoids, stilbenes, lignans, and other polyphenols with hydroxyl group(s) attached to the carbon atom on the aromatic ring (detailed classification and their food sources are listed in Table 1) [10]. It was reported that dietary consumption of polyphenols is much higher than the daily intake of several essential micronutrients, such as vitamin C, vitamin E, and carotenoids [11]. Over the past two decades, polyphenolic compounds have attracted considerable research interests because of their wide distribution in different foods and their potent antioxidant properties [11]. In addition, polyphenols were reported to modulate energy metabolism in a manner favorable for well-being and longevity and reduce the risk of aging-related chronic diseases [12,13,14,15]. Here, we provide a brief overview of the current understanding of aging and the mechanisms of cellular senescence. We propose to critically review the anti-aging effects of polyphenols and experimental evidence supporting the health-promoting effects against aging-related diseases. 

## 2. Anti-Aging Effects of Polyphenols

### 2.1. Polyphenols and Longevity

There has been great interest in identifying natural compounds for improving health, preventing chronic diseases, and extending longevity. In this regard, numerous polyphenols, including EGCG, curcumin, and quercetin, have been shown to extend lifespan in various model organisms, including *Caenorhabditis elegans (C.elegan), Drosophila Melanogaster (Drosophila)*, as well as high-fat diet-induced obese rodents [15]. Similarly, dietary resveratrol was shown to promote the health and survival of high-fat diet-induced obese male mice [16]. However, resveratrol treatment failed to extend the lifespan of chow-diet fed C57/B6 mice [17] or heterozygous mice [18,19]. In an animal study, Reutzel et al. reported that administering a mixture of six highly purified olive secoiridoid polyphenols at 50 mg /kg in the diet for six months has a long-term positive effect on cognition and brain energy metabolism in aging mice [20]. In this study, cognition was measured by behavioral tests. Brain ATP level and NADH reductase, cytochrome c oxidase, and citrate synthase mRNA expression levels are the main indicators for mitochondrial dysfunction [20]. Moreover, data from the Aging Intervention Testing Program supported by the National Institute on Aging (USA) showed that the popular anti-aging natural agents, including resveratrol, green tea extract, and curcumin did not affect lifespan of standard chow-diet-fed heterozygous mice [19]. These results suggest that these compounds may only work under certain conditions. In that regard, they may only delay or mitigate aging-associated chronic diseases in inbred mice fed a high-fat diet for a long period of time, which lack genetic diversity and are more susceptible to environment insult-induced pathological changes.

We found that dietary intake of (-)-epicatechin (EC), a flavonoid present in various foods including cocoa (*Theobroma cacao*), chocolate, berries, and tea, greatly increased survival rate in obese diabetic mice (50% and 8.4% mortality in control and EC groups after 15 weeks of treatment, respectively), whereas blood glucose levels and food intake were not modulated, suggesting that the observed effects of EC was not due to the secondary action to the changes of these variables. However, EC treatment improved skeletal muscle stress, reduced systematic inflammation, increased hepatic antioxidant glutathione concentration and superoxide dismutase activity, decreased circulating insulin-like growth factor-1, and improved AMP-activated protein kinase-α activity in the liver and skeletal muscle of diabetic mice. Therefore, EC may be a novel food-derived, anti-aging compound with direct lifespan-extending effect, given that these favorable changes are always associated with a healthier and longer lifespan [21]. Consistently, further study showed that EC (0.1–8 mmol/L) also promoted survival and increased mean lifespan of *Drosophila* [21]. To further examine whether EC indeed is able to promote health and lifespan in aged but otherwise healthy mice fed the standard chow diet, we provided 20-month old mice (equivalent of 60-65 years old in humans) with EC in drinking water (0.25% w/v) for 37 weeks, and discovered that EC-treated mice had a strikingly higher survival rate as compared with that in control group (69.7% versus 39.2%) [22]. Remarkably, the universal profiles of the serum metabolites and general mRNA expressions in skeletal muscle of old mice were shifted by EC toward those demonstrated in young mice [22]. These findings suggest that EC may be a novel anti-aging polyphenolic compound that could promote healthy lifespan in normal healthy subjects. Cocoa is particularly rich in flavonoids that include EC, flavanol-3ols, catechin, and oligomeric derivatives, making up 10% of the dry weight of cocoa powder [23,24,25,26]. Thus, it is tempting to speculate that cocoa product consumption may extend the lifespan in humans. Epidemiological studies further show that people living on the San Blas Island, who are known to consume large amounts of cocoa beverage daily, have a remarkably longer lifespan than those who live in the mainland of Panama [27,28]. Interestingly, these differences disappeared when people from San Blas island migrated to Panama city where cocoa consumption was considerably reduced [29]. Dietary chocolate intake extended average life expectancy by as much as 4 years in humans [30]. While EC only accounts for 20–30% of total flavanols in cocoa, its oligomers cannot be absorbed into circulation in rodents and humans [23,24,25]. Therefore, it is believed that the monomers might be the predominant form responsible for the beneficial effects of cocoa in vivo. Indeed, EC was found to be the dominant (>96%) flavanol in human circulation with the plasma concentration reaching more than 6 μM after ingestion of cocoa [31]. Therefore, EC may be the primary constituent in cocoa that exerts health benefits and extends lifespan in healthy subjects. 

Data from both epidemiological and large randomized clinical studies have demonstrated that Mediterranean diet (MeD) eating pattern was associated with lower risk of major chronic diseases (heart disease, cognitive decline and breast cancer), better quality of life, and longer life expectancy [32]. A 50-year follow-up study found that MeD eating habit contributes 4.4 years longer of life expectancy in Italy [33], which is similar to other two intervention studies showing that MeD is inversely associated with total mortality in Sweden [34] and the USA [35]. While Mediterranean dietary pattern is characterized by consumption of plant-based foods and olive oil, which may collectively exert health-promoting effects, emerging evidence shows that some polyphenols present in olive and olive oil likely contribute to some observed health benefits of Mediterranean diet consumption. Indeed, olive-derived polyphenols, including hydroxytyrosol, tyrosol, oleuropein, and pinoresionol [36,37], were reported to improve aging-related dysfunctions of the brain [38], and heart [39] in mice. Long-term (10 months) dietary intake of extra-virgin olive oil (10% wt/wt dry diet) rich in phenols (total polyphenol dose/day, 6 mg/kg) significantly prevented aging-related impairment in motor coordination in the rotarod test in 10-month old C57BL/6J mice [38]. Dietary intake of 10% olive oil (contains 532 mg gallic acid/kg oil) for 4.5 months significantly reduced oxidative stress in the senescence-accelerated mouse prone 8 (SAMP8) mice, which was mediated by the induction of nuclear factor erythroid 2-related factor 2 (Nrf2)-dependent gene expressions [39]. These results are in line with a new report that dietary intake of extra virgin olive oil for 6 months significantly ameliorated cognition decline in a transgenic mouse model with brain amyloid plaques and neurofibrillary tangles [40]. In a study with diabetic rats, it was also shown that a 3-month treatment with virgin olive oil (0.125 mg/kg phenols per day) displayed a stronger neuroprotective ability than that of aspirin (2 mg/kg per day) [41]. Although the mortality rate was not altered by olive oil intake because of the relatively young age of the mice used in these studies (10-months or 9–10 weeks old), it is tempting to speculate that olive oil polyphenols might have the potential to extend lifespan. Actually, increasing evidence supports that olive oil phenols delay the aging process in cells, animals, and humans [42,43]. Rechtsregulat® (RR) is the fruit and vegetable extract that is rich in polyphenols, including phenolic metabolites such as protocatechuic acid (PCA) [44]. Dilberger *et al*. reported that *C. elegans* administrated with 10% RR or 780 µM PCA showed a significant increase in heat-stress resistance, median lifespan, and activity of mitochondrial respiratory chain complexes. These results suggest that both polyphenols and the PCA can enhance the lifespan and mitochondrial function [44]. The European Food Safety Authority has approved a claim that olive oil polyphenols protect against LDL oxidation at a minimal dose of 5 mg/kg/day hydroxytyrosol [45]. 

### 2.2. Polyphenols on Aging-Related Diseases

Aging is a well-established risk factor for a wide range of diseases [15], among which neurodegenerative diseases are highly prevalent since the central nervous system (CNS) is vulnerable to aging-induced cellular malfunction and degenerative damage [46]. In addition, cellular inflammatory response progressively increases with age [47], which increases the risk for developing chronic inflammation related disease such as atherosclerosis in the elderly [48]. Furthermore, type 2 diabetes (T2D), characterized by insulin resistance and pancreatic β-cell dysfunction, may lead to a complex complications such as vascular tissue damage, depression, cognitive decline, and dementia which are tightly associated with aging [13]. Below we will review studies, mostly in vivo studies, exploring the physiological role of polyphenols in aging-related disease. We aim to critically review the studies using polyphenols as a complementary and alternative medicine that benefits humans. 

#### 2.2.1. Polyphenols in Neurodegenerative Diseases

Aging is a major risk factor for developing neurodegenerative diseases, and presently, there is no effective treatment for aging-related neurodegeneration [49]. Neurodegenerative diseases, such as Alzheimer’s disease (AD), Parkinson’s disease (PD), Huntington’s disease (HD), and amyotrophic lateral sclerosis (ALS), are characterized by the progressive loss of functional neuron cells [50]. The brain requires comparably high amount of oxygen [51] and the impaired oxidative balance in the brain has been substantially determined as the common feature of neurodegenerative diseases [52,53]. Mounting research have reported the utilization of natural polyphenols on neurodegenerative diseases due to their antioxidant and anti-aging properties [54,55,56,57,58]. Grape is an evident example of fruits that contain a high content of polyphenols [59]. Dietary supplement of grape polyphenol concentrate at 1.5 mL/kg significantly enhanced memory reconsolidation in transgenic mouse model of PD [60]. Curcumin is a polyphenol present in turmeric (*Curcuma longa*) root, which is a yellow pigment used in curry. In an old aluminum-induced neurotoxical rat model, oral gavage of 30 mg/ml/kg curcumin treatment significantly suppressed the activities of superoxide dismutase (SOD), glutathione peroxidase (GPx), glutathione-s-transferase (GST), protein kinase C (PKC), and Na^+^, K^+^-ATPase, thereby protecting the brain against aluminum toxicity in aging rats [61]. Supplementation of EC (10 mg/kg) per day for two weeks significantly ameliorated 6-hydroxydopamine (OHDA)-lesioned PD behavior, including increased locomotor activity and decreased rotational behavior, as compared with sham operated control group in male rats, suggesting a neuroprotective property of this compound [62]. Sugarcane (*Saccharum officinarum* L.) top ethanolic extract was reported to contain several polyphenols with strong antioxidant effects, including 3-caffeoylquinic acid (CQA), 5-CQA, 3-O-feruloylquinic acid (FQA), and isoorientin [63]. CQA provided at 5 mg/kg for 30 days was reported to enhance brain function of SAMP8 mice as demonstrated in the Morris water maze test [64]. Consistently, microarray analysis showed that sugarcane top ethanolic extract regulated neuron development-associated genes in the cerebral cortex of SAMP8 mice [63]. Isoorientin is a naturally occurring *C*-glycosyl flavone and exists in several dietary plants, including corn silks [65], rooibos tea [66], and buckwheat [67]. Interestingly, although isoorientin showed lower antimycobacterial activity in virulent strain [68], it remarkably protected human neuroblastoma SH-SY5Y cells from 6-OHDA-induced toxicity via the AMPK/Akt/Nrf2 signaling pathway [69]. Olive polyphenols are components of the Mediterranean diet. Grewal et al. examined the effects of purified olive secoiridoid derivatives and their metabolites on mitochondrial function in SH-SY5Y-APP695 cells, a cellular model of early AD [70]. An aging mouse model (Female NMRI mice, aged 12 months) was used to further examine the effects of purified secoiridoids (oleocanthal and ligstroside) in vivo. Results of in vitro studies showed that purified ligstroside protected against mitochondrial dysfunction by restoring ATP levels in models of early AD [70]. Female NMRI mice supplemented with oleocanthal or ligstroside (6.25 mg/kg b.w) for six months showed improved spatial working memory and ATP levels in the brain [70]. These results indicate that ligstroside can expand the lifespan in aging mice and enhance cognitive function. The ortho-diphenol hydroxytyrosol (HT) is one of the main components of extra virgin olive oil, which is an important part of the Mediterranean diet. Schaffer et al. reported that HT-rich olive mill wastewater extract (HT-E) and its main constituent HT were found to protect against ferrous iron or sodium nitroprusside-induced cytotoxicity in PC 12 neuronal cells [71]. The same group further demonstrated that HT-E also protects isolated brain cells against oxidative stress after subchronic oral administration of the extract to mice, suggesting the neuroprotective potential of HT-E [72]. The flavonoid 7,8-dihydroxyflavone (7,8-DHF) is one of the polyphenolic compounds and is naturally present in *Godmania aesculifolia*, *Tridax procumbens,* and in leaves of Primroses [73]. Rho proteins, including the small GTPase Rac1 are essential regulators of neuronal synaptic plasticity. Interestingly, the levels of Rac1 and Rab3A were restored in membrane isolated from brains of aged mice when treated with 7,8-DHF at 100 mg /kg body weight once daily for a total of 21 days via oral gavage [73]. Quercetin is one of the polyphenols found in many foods and vegetables, such as red wine, green tea, apples, and berries. Quercetin at 1 µM was found to prevent glucose-induced lifespan reduction of *C. elegans mev-1* mutants [74]. The results of this study further demonstrated that the sirtuin SIR-2.1, the nuclear hormone receptor DAF-12, and MDT-15 are essential for the effect of quercetin [74]. 

Although the etiology of AD is still elusive, the production of amyloid β (Aβ) peptides via β-secretase and γ-secretase cleavage are the pathological hallmarks of AD [75]. In addition, abnormal phosphorylation and aggregation of tau protein in neuronal cells also lead to the progression of AD [76]. As mentioned above, isoorientin is a naturally occurring polyphenol with strong antioxidant property and it has been demonstrated to protect human neuroblastoma SH-SY5Y cells from β-amyloid induced tau hyperphosphorylation, thereby exerting neuroprotective potential [65]. Treatment with green tea polyphenols significantly protected primary rat prefrontal cortical neuron cells from Aβ-induced neurotoxicity via the protein kinase B (PKB, also known as Akt) signaling pathway [77]. In addition to green tea, coffee is also a popular beverage worldwide. Epidemiological studies have shown the lifespan-extending benefits of coffee-consuming habits [78] and the potential of coffee consumption in alleviating the severity of PD [79]. Chlorogenic acid largely presents in both caffeinated and decaffeinated coffee, with a potent free radical scavenging activity and antioxidant effects [80]. Treatment of chlorogenic acid (40 mg/kg bw) significantly improved spatial memory in the Morris water maize test and alleviated neuron damage in an AD mouse model [81]. The biosynthetic compound 3’-*O*-methyl-epicatechin-5-*O*-*β*-glucuronide, which is a grape derived polyphenol, improved basal synaptic transmission and cognitive functions in a mouse model of AD [82]. Catechins are a group of bioactive polyphenols found in green tea leaves, among which (+)-catechin, EC, and (-)-epigallocatechin gallate (EGCG) are the most abundant and well-known compounds [83]. These polyphenols also can be found in other fruits and vegetables such as cocoa beans and berries [84]. In a mice model with an early appearance of learning and memory decline and an increased production of Aβ peptides, 0.05–0.1% catechin treatment decreased Aβ peptides and enhanced mice behavior in spatial learning and memory capacity tests [85]. As aforementioned, resveratrol was quite a hot topic among scientists because of the “French Paradox” a decade ago. A comprehensive review paper summarized that resveratrol extends model organisms life expectancy by up to 60% depending on the dosage, gender, genetic background, and diet composition [86]. In an intracerebroventricular injection of streptozotocin-induced brain insulin resistant rat model, 30 mg/kg/day resveratrol treatment significantly increased brain Sirt1 activity, reversed the hyperphosphorylation of tau, and enhanced the cognitive capability as compared with age-matched control rats [87]. The pathophysiology of neurodegenerative diseases has a complex mechanism and as the elderly population gets larger, aging-related neurodegenerative diseases become a major worldwide public health concern. The available in vivo and in vitro studies indicate a possible role of polyphenols in improving cognitive function and neurodegenerative diseases, future clinical trials are needed to assess the possibility using natural polyphenols as a therapeutic strategy.

#### 2.2.2. Polyphenols in Aging-Related Inflammatory Diseases

Despite the enormous complexity of aging, one of the key features is chronic inflammation [88]. A recent retrospective study testing the effects of resveratrol in AD patients (aged 50 and above) reported that 52 weeks of resveratrol treatment (1 g by mouth twice daily) significantly decreased levels of plasma inflammatory markers and induced the adaptive immune response, suggesting a promising role of resveratrol against inflammation [89]. Nuclear factor kappa light chain enhancer of activated B cells (NF-κB) is a ubiquitous transcription factor, which can be activated by acetylation and phosphorylation, thereby initiates the transcriptional cascade of a large variety of target genes that are involved in inflammation and innate immunity in mammalian cells [90]. EGCG, the major polyphenol found in green tea, was reported to extend rat median lifespan from 92.5 weeks to 105 weeks and significantly decrease the mRNA and protein levels of NF-κB, improving aging-induced oxidative status in rat liver and kidney [91]. In a randomized controlled clinical trial, consumption of turmeric extracts curcumin at 1.5 g/day for four weeks alleviated pain and swelling with minimal side effects in osteoarthritis patients (aged 50 and above) [92]. Similarly, quercetin, one of the most studied polyphenol found in various fruits [93], efficiently reduced the severity and sickness period in older patients with upper respiratory tract infection at a dose of 1 g per day for 12 weeks [94]. Treatment of isoorientin effectively inhibited the release of inflammatory cytokines in high fructose-fed mice [95]. Recently, we reported for the first time that curcumin (1 μM) and luteolin (0.5 μM) synergistically (combination index is 0.60) inhibited TNF-α-induced monocyte adhesion to human EA.hy926 endothelial cells while the individual chemicals did not have such effect at the selected concentrations [96]. Collectively, studies in both humans and animals have shown the potential of polyphenols in modulating aging-related inflammatory disorders and further analyses of the detailed mechanisms can be useful to guide clinicians and health care providers to consider polyphenols in their approach to intervene aging-associated inflammatory responses.

## 3. Potential Anti-Aging Mechanisms

### 3.1. The Antioxidant Effects of Polyphenols

According to the free radical theory, aging results from chronic imbalance (extra amount of ROS) between ROS and antioxidants, also called oxidative stress, which leads to cellular senescence, functional alterations, and pathological conditions [97,98] as discussed above. It has been well acknowledged that many polyphenolic compounds possess antioxidant properties. As exogenous antioxidants, polyphenols can combat ROS by at least four mechanisms as highlighted below. 

First, polyphenols can directly scavenge ROS because of the presence of phenolic hydroxyl groups on their molecules. The ROS-scavenging capacity of polyphenols depends on the number and position of the hydroxyl group and substituent patterns, as well as glycosylation of phytochemical molecules [99,100,101]. For example, kaempherol-3,7,4’-trimethylether, kaempherol-3,4’-dimethylether, kaempherol-7-neohesperidoside, and kaempherol, which have 1 (in the 5-position), 2, 3, and 4 hydroxyl substitutions, respectively, are 0, 1.0, 1.6, and 2.7 times of trolox equivalent antioxidant activity, respectively [102]. These data suggest that phenolic compounds with more hydroxyl groups may have a stronger antioxidant capacity. Moreover, substitution patterns in the B-ring and A-ring as well as the 2, 3-double bond (unsaturation) and the 4-oxo group in the C-ring are important for the antioxidant capacity of the compounds [103,104]. Polyphenol with a 3′,4′-o-dihydroxyl group in the B-ring, a 2,3-double bond combined with a 4-keto group in the C-ring, and a 3-hydroxyl group had the highest antioxidant activity [103,105]. Flavanols with a galloyl moiety had higher antioxidant activity than those without, and a B-ring 3′,4′,5′-trihydroxyl group further improved their efficacy [103,105]. When C-3’, 4’ positions in B ring of flavonoids are replaced by hydroxyl groups, the antioxidant activity improved remarkably, however, the numbers and substitutional positions of methoxyl and glycosyl seemed to have little effect on the antioxidant activity [106]. In an animal study, C57BL/6J mice were received a polyphenol-rich grape skin extract (PGE) diet at a dose of 200 mg/kg body weight (BW)/d for a 3-week, 6-month, and life span [107]. The results of this study showed that lifelong PGE feeding resulted in a transient, but significant change in the survival curve, although it did not affect the overall survival rate of the animals [107]. These effects of PGE are associated with enhanced signal pathways involved in energy homeostasis, antioxidant defense, and mitochondrial biogenesis, including SIRT stimulation [107].

Second, polyphenols can exert antioxidant activity through regulating endogenous antioxidant and oxidase enzyme production and activity. As a primary mechanism neutralizing oxidants, two intracellular enzymes, sodium oxide dismutase 1 (SOD1) in cytosol, and SOD2 in the matrix of mitochondria, quickly convert superoxide to hydrogen peroxide. Hydrogen peroxide is further deactivated by catalase (CAT) or glutathione peroxidases (GSH-Px) to water and oxygen. Numerous studies reported that curcumin (8 mg/kg) [108], EGCG (100 mg/kg) [109], or quercetin ( 0.027% in the diet) [110] reversed oxidative stress-caused reduction of GSH and SOD levels in mice or rats. Resveratrol protects against oxidative damage by increasing expressions of SOD1, CAT, and heme oxygenase-1 (HO-1) as well as the activity of SOD [111,112]. Similarly, oral administration of epimedium flavonoids improved CAT and GSH-Px activities by 13.58% and 5.18%, respectively, in D. melanogaster [113]. Genistein dose-dependently increased GSH-Px in breast cancer cells [114]. Flavonoid chrysin and its derivatives exhibit a high selectivity of GSH efflux (transporting intracellular GSH out of the cell), which can be used to kill chemoresistant cancer cells [115].

Third, polyphenols may enhance cellular antioxidant activity via regulating Nrf2-mediated pathway. Nrf2 is a transcriptional factor regulating the expression of several detoxifying enzymes, including SOD, GPx1, GSH, NADP(H) quinone oxidoreductase 1 (NQO1), GST, and HO-1, by binding to the antioxidant-response elements (AREs) in the promoter regions of the genes of these enzymes [116]. Numerous polyphenols, including EGCG [117], luteolin [118], curcumin [119], and epicatechin [120] can enhance Nrf2 DNA-binding activity or protein expression and subsequently increase NQO1, HO-1 and SOD expression. Resveratrol (25–50 μM) increased 2.5-fold NQO1 protein levels and a 3- to 5-fold NQO1 enzymatic activity in human k562 cells [121]. The possible molecular mechanism is that resveratrol disrupts the Nrf2-Keapl complex in the cytosol, which stimulates the translocation of Nrf2 to the nucleus where it locates the ARE-containing 5’-promoter region of NQO1, leading to its transcriptional activation [121]. 

Lastly, there is emerging evidence showing that polyphenols may counteract ROS via regulating mircoRNAs (miRNA, see more details in the next section). MicroRNAs (miRNAs) are endogenous, noncoding, single-stranded, and short (19-22 of 22 nucleotides) RNAs. miRNAs sequence-specifically bind to the 3′UTR of mRNA to repress or induce translation to regulate diverse biological pathways and processes, including cell death and proliferation, human diseases like cancer and aging. Up to now, more than 38,589 miRNAs have been cataloged in miRbase (http://www.mirbase.org, accessed on 1 December 2020), and nearly 60% of all human transcripts are predicted to be regulated by miRNAs [122]. It was recently found that some polyphenols, including quercetin, hesperidin, naringenin, anthocyanin, catechin, and curcumin reversed ApoE mutant-induced changes of miRNAs, including mmu-miR-291b-5p, mmu-miR-296-5p, mmu-miR-30c-1, mmu-miR-467b, and mmu-miR-374, which collectively regulate 34 common pathways, including the pathway of GSH metabolism [123]. Another study found that curcumin downregulated the expression of miR-17-5p, miR-20A, and miR-27a, which was shown to modulate ROS production [124]. Dietary quercetin supplementation (2 mg/g diet, 6 weeks) increased the expression levels of hepatic miR-122 and miR-125b in high fat diet-induced obese mice [125], which were associated with redox factor 1, a modulator of oxidative stress [126]. Therefore, as shown in Figure 1, polyphenols may protect cells from oxidative stress by multiple mechanisms. Indeed, it was shown that curcumin could directly scavenge ROS [127], increase the expression of endogenous antioxidants [108], activate the Nrf2 pathway [119], and modulate miRNAs [128].

### 3.2. Polyphenols and Cellular Senescence

#### 3.2.1. Cellular Senescence and Aging

Aging, happening to each living creature, is a complicated degenerative process. One of the earliest written records of human pursuits for anti-aging drugs can be found in the earliest Chinese pharmacy monograph, Shennong Materia Medica, in B. C. 220. Indeed, scientists constantly sought to decipher the driving forces behind aging and approximately 300 theories of aging have been proposed, however, there is no dominant one that has been generally accepted by the scientific community to convincingly explain the aging process [129]. Among the early theories of aging, the activity theory defines aging as the maintenance of activities and attitudes of the young and middle ages as long as possible [130]. The evolutionary theory indicates that aging is not driven by damage, but causes organelle damage and functional decline [131]. Getting old is a developmental program, which never stops. The free radical theory proposes that aging is the result of cumulative damage to DNA, proteins, lipids, and other macromolecules caused by un-neutralized free radicals [132]. In fact, there is a dynamic balance between oxidants (ROS, and reactive nitrogen species, RNS) and antioxidants in the body. ROS, primarily superoxide anion (O^2–^˙), are primarily produced by mitochondrion during energy production (about 2% of total oxygen consumption) [133]. Superoxide is quickly converted to hydrogen peroxide by two intracellular enzymes, SOD1 in cytosol and SOD2 in the matrix of mitochondria. Hydrogen peroxide is further converted into water and oxygen by catalase or GPx [134]. Endogenous antioxidant GSH and exogenous antioxidants, including vitamins C and E, dietary polyphenols are also important ROS scavengers. Chronic imbalance (extra amount of ROS) between ROS and antioxidants leads to cellular senescence, functional alterations, and pathological conditions [97,98]. 

Senescence (from the Latin word “senex“, meaning growing old), or cellular aging, is an irreversible cell-cycle arrest in the G1 phase, elicited by excessive intracellular or extracellular stress or damage [135,136]. Senescence is necessary to restrict the replication of old and damaged cells and other detrimental alterations, thereby inactivating potential malignant transformation [135,137]. Based on the kinetics of cell senescent processes, cellular senescence can be primarily categorized as acute (transient) or chronic (persistent) senescence. While acute senescence is the part of normal biological processes necessary for maintaining physiological homeostasis, and has beneficial effect within tissues during embryonic development, wound healing, or tissue repair, chronic senescence has detrimental effects within cells and tissues, particularly in the elderly because these cells and tissues are not able to clean damaged cells through autophagic process, and leads to aging and aging-associated diseases such as cancer [138]. Increasing evidence shows that senescent cells accumulate in tissues of humans, primates, and rodents with age [139], and accumulation of senescent cells was also associated with aging-related diseases such as diabetes [140], atherosclerosis [141], and obesity [142]. Interestingly, it was shown that oxidative stress is one of the major inducers of cell senescence [143,144]. Therefore, it is tempting to speculate that polyphenols as exogenous antioxidants may have the potential to prevent cellular senescence and thereby the aging process.

#### 3.2.2. The Effects of Polyphenols on Cellular Senescence

Polyphenols treatment has beneficial effects over certain types of disease due to their action on preventing cellular senescence. A combination treatment with a senolytic drug Dasatinib and quercetin, a well-studied flavonol present in many plants, reduced the accumulation of senescent cells in adipose tissue by suppressing the senescence-associated β-galactosidase activity [145]. In agreement with this in vitro finding, the combination of Dasatinib and quercetin alleviated senescence-related idiopathic lung fibrosis [146]. In a senescence mouse model SAMP8 mice, the group consuming a diet with 532 mg/kg olive oil phenols for 4.5 months showed significantly lower levels of oxidative damage in the heart and induced longevity-related genes expression as compared to the group consuming diet with only 44 mg/kg olive oil polyphenols [39]. Consistently, it was shown that chronic treatment of pre-senescent human lung and neonatal human dermal fibroblasts with 1 μM hydroxytyrosol or 10 μM oleuropein aglycone effectively reduced senescent cell numbers as demonstrated by measuring β-galactosidase-positive cell numbers and p16 protein expression [147]. In line with this finding, oleuropein treatment delayed the appearance of senescence morphology and extended the life span of human embryonic fibroblasts IMR90 and WI38 cells by approximately 15% [148]. Gallic acid was reported to suppress β-galactosidase activity and the expression of oxidative stress markers in rat embryonic fibroblast cells [149]. These results suggest that polyphenols may be able to modulate cellular senescence thereby influencing aging process.

### 3.3. Polyphenols May Exert Anti-Aging Effects by Targeting Microrna

MicroRNAs (miRNAs) are small non-coding RNA molecule secreted from cells into peripheral body fluids, including blood, saliva, and urine, either in association with RNA-binding proteins like Argonaute 2, or bound to high-density lipoproteins, and some ‘circulatory’ miRNAs have been proposed as noninvasive biomarkers of aging [150]. Numerous miRNAs have been shown to directly affect lifespan as demonstrated by overexpressing or knockdown of the miRNA in *C. elegans, Drosophila*, and mice. These miRNAs, such as miR-125, miR-17, let-7, AGO1, and AGO2, regulated well-known aging signaling pathways, including target of rapamycin (TOR), insulin/insulin-like growth-factor (IGF-1) signaling, sirtuins deacetylases, mitochondrial/ROS signaling, and DNA-damage response [151]. In addition, using cells or mouse models, a number of miRNAs, including miR-1000, miR-455-3p, miRNA-17/20a, and miR-34a, have been shown to extend longevity through improving aging-caused dysfunctions of organs such as the brain [152], muscle [153], bone [154], and heart [155], respectively. However, most studies exploring and identifying lifespan-modulating miRNAs used *C. elegans and Drosophila,* and miR-17 is the only miRNA that has been reported to directly extend lifespan in mice [156].

While studies investigating the effects of polyphenols on miRNA expression are limited, which is an intriguing area to explore in the future, emerging evidence shows that dietary intake of some polyphenols modulates the expression of miRNAs that are involved in longevity. One study found that a low dose of supplementation of quercetin, hesperidin, naringenin, anthocyanin, catechin, or curcumin (0.006%, w/w, two weeks) in the diet modulated the expression of a broad range of miRNAs and rectified the ApoE-mutantation-induced changes of miRNAs in the livers of ApoE-deficient mice [123]. These polyphenol-modulated miRNAs, including mmu-miR-291b-5p, mmu-miR-296-5p, mmu-miR-30c-1, mmu-miR-467b, and mmu-miR-374, regulate 30 common pathways, including MAPK signaling pathway, the calcium signaling pathway, the insulin signaling pathway, as well as oxidative phosphorylation, some of which are involved in longevity in mice [123]. In addition, miR-17, a mammalian longevity miRNA that directly targets insulin receptor substrate (Irs1) and adenylate cyclase 5 (Adcy5), can be regulated by polyphenols. For instance, catechin, proanthocyanins, naringin [123] and genistein [157] upregulated miR-17 expression in mice. Similarly, the expression of let-7, a common longevity miRNA conserved across *C. elegans, Drosophila*, mouse, and humans, was enhanced by catechin, proanthocyanins, and naringin in ApoE mice [123]. Therefore, some polyphenols may act through miRNAs to regulate aging-related pathways, suppress inflammation and ROS production, and improve lipid metabolism, leading to a healthier and extended lifespan [128]. The effect of polyphenols on the expression of miRNAs may not be specific. For instance, epicatechin was shown to modulate more than 73 miRNAs involved in various cellular functions in human endothelial cells [158]. In diabetic patients, dietary intake of grape extract (8.1-16.2 mg polyphenols) upregulated miR-21, miR-181b, miR-663, and miR-30c concomitted with the lower levels of inflammatory cytokines such as IL-6, chemokine ligand 3, IL-1, and TNF-α [159]. However, it is unclear whether any of these miRNA directly mediates the anti-inflammatory action of the grape extract.

### 3.4. Polyphenols and NO Bioavailability

Endothelial dysfunction, resulting from inflammation, obesity, diabetes, hypertension, hyperlipidemia, and other related metabolic syndromes, is a major pathogenic cause of cardiovascular disease [160]. Endothelial dysfunction impairs the production and bioavailability of endothelial nitric oxide (NO) synthase (eNOS)-derived NO, which is the key regulator of vascular tone, blood pressure, and vascular inflammation [161]. The endothelial production and bioavailability of NO also progressively decreases with aging [162], which is at least partially ascribed to the increased production of ROS [163,164]. In normal condition, eNOS is coupled to generate NO from oxidation of L-arginine. However, excess oxidative stress causes the oxidation of tetrahydrobiopterin, a critical cofactor for eNOS, which leads to eNOS uncoupling from producing NO, but diversion to reduce oxygen to form superoxide [165,166], thereby reducing the bioavailability of NO that subsequently accelerates the development of vascular disease [163,164]. Thus, promoting the eNOS expression/activity and/or NO bioavailability would be effective methods to alleviate aging-associated endothelial dysfunction and subsequently delay the development of cardiovascular disease. Plenty of studies have demonstrated that polyphenolic compounds have protective actions over cardiometabolic syndrome [167], among which eNOS expression/activity and NO bioavailability are the most determined mechanisms [161,168,169,170]. Morin, a flavonoid, can effectively protect human ventricular myocytes, saphenous vein endothelial cells, and erythrocytes against oxyradicals-induced damage [171]. Further, morin treatment promoted eNOS-mediated NO production and vasodilation of aorta in STZ-induced diabetic mice by activating the Akt signaling pathway [172,173]. As aforementioned, resveratrol has attracted mounting research interests [174,175]. Resveratrol treatment increased the eNOS transcriptional activity and the eNOS-derived NO production in human umbilical vein endothelial cells [176]. Protocatechuic acid (PCA) is a major metabolite of green tea polyphenols with a strong antioxidant property [177]. Administration of PCA (200 mg/kg/day) significantly improved insulin- and IGF-1-induced vasorelaxation in aging spontaneously hypertensive rats via activating the PI3K/NOS/NO pathway [178]. Cyanidin-3-glucoside (Cy3G), a typical anthocyanin found in deep-colored plants [179], has been shown to promote eNOS protein expression and subsequently increase NO output in bovine artery endothelial cells [170]. Interestingly, multiple polyphenols, including catechin, oleuropein, quercetin, and EGCG, were found to reduce nitrite to NO in the stomach, suggesting that polyphenols may be a nitrite reductant due to the hydroxyl groups on the phenol ring [180]. Furthermore, 12 weeks of curcumin supplementation improved resistance artery endothelial function by increasing vascular NO bioavailability and reducing oxidative stress [181]. In diabetic rats, treatment with green tea extract, which is primarily composed of EGCG, ameliorated the diabetes-induced reduction of tetrahydrobiopterin, uncoupling of eNOS, and thus increased NO bioavailability and reduced oxidative stress [165].

### 3.5. Polyphenols May Promote Mitochondrial Function

In addition to the direct action on eNOS expression/activity, polyphenols were reported to activate Sirt1 [182,183], which is an upstream regulator of eNOS [184,185], therefore Sirt1-mediated mitochondrial biogenesis might underlie the anti-aging actions of polyphenols against oxidative stress. Indeed, resveratrol treatment increased mitochondrial biogenesis in wild-type mice but not eNOS knockout mice [179]. Activation of peroxisome proliferator-activated receptor-γ coactivator-1α (PGC-1α), the key regulator in mitochondrial biogenesis, has been reported to protect against aging-related diseases [16,185]. In vivo, resveratrol promoted liver PGC-1α activity and significantly extended the lifespan of mice fed a high-calorie diet [186]. In vitro, resveratrol treatment increased adenosine monophosphate (AMP)-activated protein kinase (AMPK) phosphorylation in CHO cells [186], suggesting the involvement of AMPK/Sirt1 signaling pathway in the action of resveratrol extending lifespan in mammals. Polyphenols treatment (resveratrol, apigenin, and S17834, a synthetic polyphenol) have been reported to phosphorylate AMPK in HepG2 cells, thereby subsequently protecting hepatocytes from high glucose-induced lipid accumulation [187]. Dysfunctional mitochondria can cause imbalanced ROS accumulation based on the free radical theory, which may exacerbate the progress of aging. Thus, targeting mitochondrial function can be an effective approach to slow aging. Interestingly, resveratrol treatment can improve the quality of oocytes from aged cows (>10 years old) by upregulating mitochondrial biogenesis [188], suggesting a potential mechanism to slow maternal aging. Hydroxytyrosol, a phenolic compound found in olive oil, effectively increased mitochondria number in 7PA2 cells, a well-established cell model to study AD [189], suggesting polyphenols may help alleviating the energetic deficit of AD patients. 

## 4. Conclusions and Perspectives

Interest in the use of complementary and alternative medicine, particularly polyphenol-rich natural products, has increased considerably to improve the health and well-being of humans for the past two decades. Some polyphenols have even been shown to extend lifespan in various model organisms. Polyphenols at supraphysiological or higher doses are well recognized to directly scavenge ROS by donating an electron or hydrogen atom. However, they may exert antioxidant activity in vivo via other mechanisms as discussed in this paper, given their relatively poor bioavailability. Aging and aging-related disorders are complex and certainly influenced by dietary habits and genetic backgrounds. This review is rather narrow considering the tremendous efforts of researchers investigating the molecular mechanisms underlying the beneficial actions of polyphenols, however, we highlighted emerging evidence that may provide new mechanisms underlying the antioxidant and anti-aging actions of polyphenols. While the use of polyphenols in preventing aging-related disorders has proven to be promising in various model organism-based studies, the safety and potential health benefits from long-term use of the individual pure compound in humans are still uncertain. It should be noted that one food may contain at least several and even hundreds of polyphenols [190], and some diets such as Mediterranean diet have multiple polyphenol-rich foods, which collectively contain 290 different polyphenols [191,192]. Therefore, it could be misleading to attribute the benefits of consuming particular polyphenol-containing foods to the individual polyphenols, which are often given at much higher doses, particularly in animal and in vitro studies, than those possibly obtained from consuming the relevant foods or supplements by humans [193]. In the future, it may be more relevant and interesting to investigate the beneficial effects of the combination of multiple polyphenols or polyphenol-rich foods, as a combination of polyphenols may exert synergistic or additive beneficial effects [192,194].

## Figures and Tables

**Figure 1 antioxidants-10-00283-f001:**
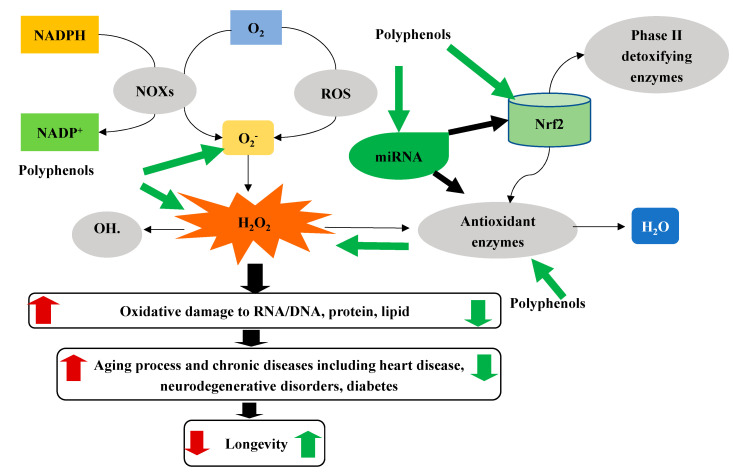
Schematic diagram of polyphenols antioxidant mechanisms. ROS, reactive oxygen species; Nrf2, nuclear factor erythroid 2-related factor 2.

**Table 1 antioxidants-10-00283-t001:** Classification of polyphenols and dietary sources.

Classification	Subgroups	Dietary Sources
Phenolic acids	Hydroxybenzoic acids	Wine, sorghum, dried date, blackberry, apple juice, olive, chicory, black tea, etc.
	Hydroxycinnamic acids	
	Hydroxyphenylacetic acids	
	Hydroxyphenylpropanoic acids	
	Hydroxyphenylpentanoic acids	
Flavonoids	Anthocyanins	Soy, bean, wine, berries, grape, plum, pomegranate juice, olive, etc.
	Chalcones	
	Dihydrochalcones	
	Dihydroflavonols	
	Flavanols	
	Flavanones	
	Flavones	
	Flavonols	
	Isoflavonoids	
Stilbenes	Resveratrol	Grape, wine, lingonberry, etc.
	Dihydroresveratrol	
	Pallidol	
	Pinosylvin	
	Piceatannol	
	Pterostilbene	
Lignans	Sesamin	Barley, apricot, peach, kiwi, beer, sesame seed, etc.
	Secoisolariciresinol	
	Isohydroxymatairesinol	
	Conidendrin	
	Episesamin	
Other polyphenols	Tyrosols	Beer, wine, olive, olive oil, turmeric, curry, etc.
	Phenolic terpenes	
	Methoxyphenols	
	Hydroxyphenylpropenes	
	Curcuminoids	
	Alkylphenols

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
