# Peer review of "Dietary Anti-Aging Polyphenols and Potential Mechanisms"

_antioxidants, 2021, doi:10.3390/antiox10020283_

Round 1
Reviewer 1 Report
This is a well-written literature review on polyphenols and their anti-aging effects. However, I have some comments that I recommend addressing before publication. Most of them are related to the fact that many studies cited by authors are quite outdated. Is this because there have not been enough studies focused on investigating polyphenols and anti-aging effects in the recent years? Which evidence found in the cited studies have been confirmed in more recent experiments or human trials?
- A high number of references are from several years ago (>8 years), and this for a literature review is not the most appropriated. I suggest authors to review some of these references and get more updated ones. For example, in first paraphrase of section 2.1. references are from <2011, are there any more recent studies showing the effects of polyphenols on longevity in animal or human studies?
- Ref 20 is missing the year, this happen with few others, please review bibliography, and make sure it is in accordance with the journal guidelines.
- Ref 21. (also missing year in bibliography) is from 2011, so reword “We recently found….”. Same for references 38 and 39, line 123.
- Line 129: reword, reference 39 is already cited above so it is not “another” study.
- Line 139: “increasing evidence” citing an article from 2012 is not appropriated. What have been confirmed from those evidence in recent years? Have there been any studies recently in humans confirming any of those findings?
- Lines 152-155 is missing reference.
- Line 175: regarding AD and polyphenols, authors should cite recent work, including this recent review: https://doi.org/10.3390/microorganisms8020199 or studies such as https://doi.org/10.1016/j.jnutbio.2018.02.001
- section 3.1. Again, I suggest reviewing more recent studies and the format of the bibliography, few of them are missing the year of publication.
- Figure 1 should include a brief description and full name for some of the less common abbreviations (ie Nfr2) represented in the figure.
- Section 3.2.1. Same as for comment in 3.1.
I suggest including some table summarizing the results from recent studies on animal models and humans trials on polyphenols effects in aging and longevity.
Author Response
We are grateful for the reviewer’s thoughtful suggestions and for the opportunity to send a revised version of the manuscript for review. We have carefully addressed each of the points raised by the reviewers as below. Further, we have thoroughly reviewed the manuscript and corrected grammatical and typographical errors. Finally, the manuscript and references are formatted according to the journal guideline. I hope that these revisions are satisfactory and the manuscript can be accepted for publication.
Comments:
This is a well-written literature review on polyphenols and their anti-aging effects. However, I have some comments that I recommend addressing before publication. Most of them are related to the fact that many studies cited by authors are quite outdated. Is this because there have not been enough studies focused on investigating polyphenols and anti-aging effects in the recent years? Which evidence found in the cited studies have been confirmed in more recent experiments or human trials?
Response: thanks for this suggestion. Overall, studies focusing on investigating the antiaging action of polyphenols are still very limited. We searched literature again and cited those as we found in the relevant areas.
A high number of references are from several years ago (>8 years), and this for a literature review is not the most appropriated. I suggest authors to review some of these references and get more updated ones. For example, in first paraphrase of section 2.1. references are from <2011, are there any more recent studies showing the effects of polyphenols on longevity in animal or human studies?
Response: thanks for this suggestion. Overall, studies focusing on investigating the antiaging action of polyphenols are still very limited. We searched literature again and cited those as we found in the relevant areas.
Ref 20 is missing the year, this happen with few others, please review bibliography, and make sure it is in accordance with the journal guidelines.
Response: I like to thank this thorough review and we reviewed the references and added the missed information and corrected formats where necessary.
Ref 21. (also missing year in bibliography) is from 2011, so reword “We recently found….”. Same for references 38 and 39, line 123.
Response: This is added, and the sentence has been reworded as suggested., Thanks.
Line 129: reword, reference 39 is already cited above so it is not “another” study.
Response: It is re-arranged and reworded as suggested.
Line 139: “increasing evidence” citing an article from 2012 is not appropriated. What have been confirmed from those evidence in recent years? Have there been any studies recently in humans confirming any of those findings?
Response: We have cited more recent publications.
Lines 152-155 is missing reference.
Response: These sentences are now deleted as this information is out of the scope of the review content.
Line 175: regarding AD and polyphenols, authors should cite recent work, including this recent review: https://doi.org/10.3390/microorganisms8020199 or studies such as https://doi.org/10.1016/j.jnutbio.2018.02.001
Response: This reference is added. Thanks.
section 3.1. Again, I suggest reviewing more recent studies and the format of the bibliography, few of them are missing the year of publication.
Response: As addressed above, we have found some more recent papers that are relevant to this review and have them added in the respective areas.
Figure 1 should include a brief description and full name for some of the less common abbreviations (ie Nfr2) represented in the figure.
Response: A brief description has been added with some abbreviations defined.
I suggest including some table summarizing the results from recent studies on animal models and humans trials on polyphenols effects in aging and longevity.
Response: Some tables summarizing the results have been added.
Reviewer 2 Report
Paper: Dietary anti-aging polyphenols and potential mechanisms described by Jing Luo et al.
The aim of this review is to critically evaluate the experimental evidence demonstrating the beneficial effects of polyphenols on aging-related diseases. Authors highlighted on the potential anti-aging mechanisms of polyphenols, including antioxidase signaling, preventing cellular senescence, targeting microRNA, influencing NO bioavailability, and promoting mitochondrial function. This paper before accepting needs some correction:
- Key words- should be different than in title
- L 46 - stilbenes, and lignans – are not belongs to polyphenols groups but other class of bioactive compounds
- L150- beta - describe as symbol
- in vivo – should be italic
- if Authors describe common name of plant i.e. cocoa or turmeric etc – should add some latin name
- Authors should add some more information in a new section about which plant as fruits, roots or herbs are valuable to anti-aging activity. Propose to add this information as table because title of this review paper suggest “dietary ….” what mean and other people want to know which plant food rich in polyphenols should eat every day. Authors are from Department of Human Nutrition, Foods and Exercise or Department of Human Sciences or Guangdong Provincial Key Laboratory of Food, Nutrition and Health, Department of Nutrition, School of Public Health therefore it’s easy to prepare this information for other readers.
- L 479 or 494: polyphenol-rich natural products – what kind of natural products;
- L 480 – Some polyphenols – which, add info
Author Response
We are grateful for the reviewer’s thoughtful suggestions and for the opportunity to send a revised version of the manuscript for review. We have carefully addressed each of the points raised by the reviewers as below. Further, we have thoroughly reviewed the manuscript and corrected grammatical and typographical errors. Finally, the manuscript and references are formatted according to the journal guideline. I hope that these revisions are satisfactory and the manuscript can be accepted for publication.
Key words- should be different than in title
Response: they are now changed and different from those in the title.
L 46 - stilbenes, and lignans – are not belongs to polyphenols groups but other class of bioactive compounds
Response: After carefully reviewing some papers, stilbenes and lignans are also categorized into polyphenols as they all have the aromatic ring with one or more hydroxyl groups on it. As suggested by another reviewer, a table demonstrating the classification of polyphenols was added.
L150- beta - describe as symbol
Response: This is changed.
in vivo – should be italic
Response: Change has been made as suggested.
if Authors describe common name of plant i.e. cocoa or turmeric etc – should add some latin name
Response: We have added the scientific names of the plants.
Authors should add some more information in a new section about which plant as fruits, roots or herbs are valuable to anti-aging activity. Propose to add this information as table because title of this review paper suggest “dietary ….” what mean and other people want to know which plant food rich in polyphenols should eat every day. Authors are from Department of Human Nutrition, Foods and Exercise or Department of Human Sciences or Guangdong Provincial Key Laboratory of Food, Nutrition and Health, Department of Nutrition, School of Public Health therefore it’s easy to prepare this information for other readers.
Response: We agree with the reviewer and such a table listing the dietary sources of polyphenols with potential anti-aging effect is added.
L 479 or 494: polyphenol-rich natural products – what kind of natural products;
Response: They include many vegetables, fruits, olive oil, whole grains, etc., which are specified in this revised manuscript.
L 480 – Some polyphenols – which, add info
Response: They are now specified.
Round 2
Reviewer 1 Report
Author have addressed most of my previous comments but not all of them. Section 3.1 for example still keep quite outdated (there are almost no references from the last decade in the whole section with the exception of a couple of them), so no significant changes in terms of references have been made here. Moreover, reference list still have some missing year for example so authors should really check this again carefully and make sure that bibliography is correct, in a review I believe this ins of special interest. Moreover, although authors answered that they added the references that I proposed, I haven’t been able to find them in the text. It is not strictly necessary to add these references as they have added others that are ok now but authors should make sure before answer that their response is in accordance with the changes made to the manuscript. Given these reasons, I cannot recommend the manuscript for publication in its current form but if authors amend these minor changes I believe that the manuscript will be suitable for publication in Antioxidants,
Author Response
Thank you very much for your time reviewing the manuscript again.
We have added more relevant references and the major findings from these papers are also summarized. Please see the track changes in the manuscript.